# SARS-CoV-2 in Pediatric Inpatient Care: Management, Clinical Presentation and Utilization of Healthcare Capacity

**DOI:** 10.3390/healthcare9091190

**Published:** 2021-09-09

**Authors:** Christine Busch, Maximilian Blickle, Beatrix Schmidt, Laura Katharina Sievers, Constanze Pfitzer

**Affiliations:** 1St. Joseph Krankenhaus Berlin-Tempelhof GmbH, 12101 Berlin, Germany; christine.busch@gmail.com (C.B.); mblickle@gmail.com (M.B.); beatrix.schmidt@sjk.de (B.S.); 2Department of Congenital Heart Disease/Pediatric Cardiology, Deutsches Herzzentrum Berlin, 13353 Berlin, Germany; constanze.pfitzer@charite.de; 3Department of Internal Medicine I, Campus Kiel, University Hospital Schleswig-Holstein, 24105 Kiel, Germany; 4Clinic of Neonatology, Charité-Universitätsmedizin Berlin, 13353 Berlin, Germany; 5Berlin Institute of Health, Charité, 10178 Berlin, Germany

**Keywords:** SARS-CoV-2, pediatric inpatient care, hospital admissions, pediatric patients

## Abstract

This study scrutinizes management and clinical presentation of severe acute respiratory syndrome coronavirus type 2 (SARS-CoV-2) in pediatric inpatient care and evaluates the utilization of pediatric healthcare capacity during the pandemic. Within this retrospective cohort study, we systematically reviewed data of all 16,785 pediatric patients (<18 years admitted to our clinical center between January 2018 and June 2021). Data on SARS-CoV-2 test numbers, hospital admissions and clinical characteristics of infected patients were collected. Since January 2020, a total of 2513 SARS-CoV-2 tests were performed. In total, 36 patients had a positive test result. In total, 25 out of 36 SARS-CoV-2 positive children showed at least mild clinical symptoms while 11 were asymptomatic. Most common clinical symptoms were fever (60%), cough (60%) and rhinitis (20%). In parallel with the rising slope of SARS-CoV-2 in spring and fall 2020, we observed a slight decrease in the number of patients admitted to the pediatric department while the median duration of hospital treatment and intensive care occupancy remained unchanged. This study underlines that SARS-CoV-2 infected children most frequently exhibit an asymptomatic or mild clinical course. Noteworthy, the number of hospital admissions went down during the pandemic. The health and economic consequences need to be discussed within health care society and politics.

## 1. Introduction

In December 2019, a pneumonia associated novel coronavirus (SARS-CoV-2) was identified in China and rapidly spread across the world causing significant morbidity and mortality [1,2]. The coronavirus infectious disease 19 (COVID-19) can cause severe respiratory illness such as acute respiratory distress syndrome and pneumonia and may also affect other organ systems such as coagulation or nervous systems [3,4,5]. Advanced age and underlying health conditions such as hypertension, diabetes or cardiovascular disease appear to be risk factors predisposing for a more severe clinical course of COVID-19. During the pandemic, the health care systems of numerous countries have faced major challenges due to the high numbers of adult patients requiring intensive care treatment. In contrast, children infected with SARS-CoV-2 most frequently exhibit an asymptomatic or mild clinical course and therefore less frequently need to be hospitalized or transferred to the intensive care unit (ICU) [6,7]. Based on the broad range of SARS-CoV-2 associated clinical symptoms including highly infectious asymptomatic patients, it has become standard in most hospitals to test patients of all age groups for SARS-CoV-2 before admission to hospital, regardless of symptoms. However, data on incidence, clinical presentation and severity of COVID-19 in hospitalized children remain insufficient, especially in Germany [8]. Further, it is assumed that the SARS-CoV-2-pandemic and the consecutive lockdowns have affected the rate of hospital admissions [9], but data pertaining to pediatric hospitalization rates has not been sufficiently studied to date. Our study aimed to gather more information and carry out analyses on the following three points: (1) The incidence of positive SARS-CoV-2 test results and symptomatic COVID-19 among hospitalized children. (2) The effect of COVID-19 on the frequency and duration of inpatient treatment. (3) The frequency and severity of clinical symptoms among SARS-CoV-2 positive children.

## 2. Materials and Methods

### 2.1. Study Design and Patient Cohort

This is a retrospective single center cohort study on pediatric patients hospitalized at the St. Joseph Krankenhaus Berlin, Germany. The ethics committee of the Charite–Universitätsmedizin Berlin, Germany, approved this study (no. EA2/066/20).

We systematically reviewed data of all 16,785 patients under the age of 18 who were admitted to St. Joseph Krankenhaus Berlin between January 2018 and June 2021. Figure 1 summarizes our patient cohort and gives an overview of the two distinct subgroups we defined according to the SARS-CoV-2-test procedure during the time of the study. Cohort 1 includes all patients hospitalized during the period of our study without any preselection criteria. This subgroup was analyzed regarding rate of hospital admissions before, during and after the onset of the SARS-CoV-2 pandemic.

The incidence and clinical course of COVID-19 were analyzed in a subgroup of patients who had received an oro- or nasopharyngeal swab for SARS-CoV-2 by polymerase chain reaction (PCR) based on clinical suspicion between March and September 2020 (Cohort 2). During this period, symptom-based screening covered 384 patients while a total of 448 tests were performed. Children from the department of pediatric psychosomatic were excluded from cohort 2 because of differing test strategies. Patients that presented themselves with at least one of the following symptoms received an oro- or nasopharyngeal swab for SARS-CoV-2 and were included in Cohort 2: fever, signs of an acute respiratory infection and/or diarrhea for infants less than six months of age. In cohort 2, the following information was collected: demographic data, clinical data (body temperature, clinical symptoms, duration of symptoms before hospitalization, comorbidities), diagnostic data (laboratory measurements, imaging: sonography and/or X-ray) and therapeutic data (duration of treatment, oxygen demand, ventilation, admission to intensive care unit, antibiotic therapy, final diagnosis). At the onset of the second COVID-19 wave in Germany, a mandatory screening via professional SARS-CoV-2 Antigen test of all patients admitted to the hospital was initiated in October 2020. An exception was made for newborns referred to the department of neonatology. All mothers giving birth were tested upon admission. However, newborns, born within the hospital and admitted to neonatology from the delivery room or maternity ward, were not tested. In contrast, newborns who came from home, were tested. Patients with a positive SARS-CoV-2-Rapid test were tested again via an oro- or nasopharyngeal swab for SARS-CoV-2 by PCR.

### 2.2. Statistical Analysis

Data was collected in an Excel^®^-database (Microsoft, Redmond, WA, USA) and visualized with Numbers (Apple Inc., Cupertino, CA, USA). Descriptive analysis was performed with presentation of total number, mean and standard deviation for normal distribution or median and minimum and maximum value for non-normal distribution. Categorical variables were reported as the percentage value of the collective. To compare the number of positive SARS-CoV-2 tests in our cohort with the overall dynamic of the pandemic, we included data on the overall SARS-CoV-2 incidence in Germany of the Robert-Koch-Institute (RKI) in Figure 2 [10].

## 3. Results

In total, 16,785 pediatric patients were enrolled in our study, among these 6108 patients after the beginning of the SARS-CoV-2 pandemic in January 2020 (Table 1). During this time, a total of 2,513 SARS-CoV-2 tests were performed. At the beginning of the pandemic, SARS-CoV-2 testing was carried out based on clinical suspicion, e.g., in patients with fever and/or respiratory symptoms such as cough, rhinitis and sore throat. Starting in October 2020, all patients admitted to the hospital were tested for SARS-CoV-2 at least once. An exception was made for newborns referred to the department of neonatology. 

At our study site SARS-CoV-2 test numbers gradually increased during the course of the pandemic (Table 1). After implementation of the mandatory screening in October 2020, all patients were tested with at least an antigen test. In total 36 patients tested positive for SARS-CoV-2 with a PCR test. The test positive rate in our patient group was lower than the overall test positive rate in Germany. In total, 25 of the 36 SARS-CoV-2 positive children showed at least mild clinical symptoms, while 11 remained asymptomatic (Table 1). The incidence of SARS-CoV-2 in Berlin, Germany, where our study was performed, mimicked the nationwide course with three peaks until summer 2021 (Figure 2A). In total, 15.5% of the infected in the city of Berlin were children (Median weekly fraction 15.6%). All age groups were affected. The proportion of SARS-CoV-2 positive children in all positive tests varied between 5% and 34% while children account for roughly 17% of the population of the city. Positive SARS-CoV-2 PCR results in our study population clustered with increased overall incidence but were not limited to that (Figure 2A). Parallel to the rising number of SARS-CoV-2 cases as depicted in Figure 2A, we observed a slight decrease in the number of patients admitted to the pediatric department in the months of March, April, May and again in the months of October, November and December 2020 (Table 1, Figure 2B) compared to the same months of the previous years. The percentage of children admitted to the ICU during the pandemic remained relatively stable with a median of 20.3% (Figure 2B). 

The distribution of all admitted patients between the different pediatric subspecialties during our study was as follows: 60.1% general pediatrics, 24.1% neonatology, 14.8% pediatric surgery and 1.1% pediatric psychosomatics. The number of patients treated in the departments of neonatology and pediatric psychosomatics remained stable during the study whereas patient numbers in pediatric surgery and general pediatrics went down markedly during the peak months of SARS-CoV-2 incidence.

The median hospital stay duration remained at 2 days throughout the entire period of time observed in our study. 

Up to date there have only been few reported cases of SARS-CoV-2 infections with severe outcome in children [11]. The majority of children and adolescents only suffer from mild symptoms and previous studies have shown that 15–17% of cases remain asymptomatic [12]. In our study, around one-third of all patients remained asymptomatic while two thirds showed at least mild symptoms. During the phase of symptom-based screening (March 2020–September 2020) only 1.3% of tests were positive leading to a SARS-CoV-2 incidence of 0.2% among all admitted patients. During the period of general screening between October 2020 and June 2021, 1.1% of all patients tested positive among these 11 asymptomatic patients tested positive. Our findings are comparable to those of other studies performed at similar incidence levels.

In order to analyze how to better differentiate symptomatic SARS-CoV-2 patients from other infectious diseases we more closely examined a subgroup (cohort 2) that was defined by suspicious symptoms such as fever and/or upper respiratory symptoms or diarrhea in infants under the age of 6 months.

All 384 symptomatic patients who were admitted to the hospital between March and October 2020 and had received an oropharyngeal swab for a SARS-CoV-2 PCR test were included (Table 2). Among these, 223 patients were male and 161 were female with a median age of 19 months. A total of 1.8% of patients tested positive (6 of 384 patients). In total, 122 of the patients had underlying medical conditions such as neurological, psychiatric, pulmonary, cardiac, urogenital or syndromal comorbidities or had a history of preterm birth. Only one of the SARS-CoV-2 positive children had a comorbidity, namely a history of preterm birth >30 weeks of gestation (Table 2). Weight and height of the SARS-CoV-2 positive children were distributed along all percentiles; however, their number was too low to analyze whether hypo- or hypertrophic children were overrepresented among the infected (Table 2). Extending the analysis to all 36 SARS-CoV-2 positive children in Cohort 1, 20 male and 16 female patients were identified. Almost half of the patients were younger than 6 months (17/36), 6 patients were aged 7–24 months, 4 patients aged 2–12 years and 9 patients were older than 12 years. In total, 9 out of the total of 36 SARS-CoV-2 positive children had an underlying medical condition: 2 had psychiatric and 2 urogenital comorbidities, 2 were adipose, 1 suffered from hemophilia, 1 had a history of preterm birth <30 weeks of gestation and 1 had a history of preterm birth >30 weeks of gestation.

During the period of symptom- based SARS-CoV-2 screening we observed that fever, cough, dyspnea or tachypnea and presence of respiratory symptoms for more than 2 weeks were frequently reported among SARS-CoV-2 positive pediatric patients (Table 3, Figure 3). 60.0% of SARS-CoV-2 positive children in Cohort 2 suffered from fever but only 1.2% of the patients with fever had a positive SARS-CoV-2 test (Table 3). We then analyzed the clinical presentation of SARS-CoV-2 positive children during unbiased screening of all patients admitted to our clinic and found that 11% were completely asymptomatic and 67% of symptomatic patients suffered from fever (Figure 3).

Abdominal symptoms or neurological abnormalities were not observed in SARS-CoV-2 positive patients except for one fever-induced seizure (Table 3). The most frequent final diagnoses for admittance of SARS-CoV-2 negative patients were upper airway infections, obstructive bronchitis, pneumonia or newborn infection without further specification (Table 4). More than 10% of admitted patients had no infectious primary diagnosis.

Further, the clinical examination details and laboratory results were analyzed. Neither SARS-CoV-2 negative nor SARS-CoV-2 positive children showed an association of febrile, subfebrile or normal body temperature with the assumed site of infection. In both groups, fever and respiratory symptoms was not an appropriate parameter to distinguish between pneumonia and upper airway infection.

Laboratory results of CRP, leukocyte count, thrombocyte count, hemoglobin, fractional lymphocytes and fractional neutrophils revealed a tendency towards lower CRP and increased lymphocyte vs. neutrophil ratio in SARS-CoV-2 positive patients (data not shown).

Hospitalized SARS-CoV-2 pediatric patients did not need any form of therapy escalation compared to children tested negative in cohort 2. None of the SARS-CoV-2 positive but 12% of all patients from cohort 2 required intensive care treatment (5% with respiratory support such as CPAP, nasal high flow or invasive ventilation). In total, 21% of SARS-CoV-2 positive and symptomatic and 16% of the other children required oxygen. Bronchodilator medication was administered to 40% and empiric antibiotic treatment to 20% of the patients that had tested positive for SARS-CoV-2. Only 32% of patients that had tested negative for SARS-CoV-2 required bronchodilator medication but 36.2% received empiric antibiotic treatment. 

The median hospital stay duration was 2 days for patients with an upper airway infection (equal for SARS-CoV-2 negative and positive) and 4 days for patients with pneumonia regardless of the infectious agent (SARS-CoV-2 vs. other specific agents or not identified; data not shown). 

## 4. Discussion

Our study provides insight into incidence, clinical presentation and management of SARS-CoV-2 in pediatric patients.

SARS-CoV-2 detection rates in our cohorts did not always follow the detection rate fluctuations observed in the general population (Figure 2A). The infection rates among children may be influenced by various external factors such as political issues like home office for parents, hygiene measures in daycare and schools, infectiosity of virus variants for the young, or local outbreaks at schools.

During the pandemic the total amount of children requiring inpatient treatment appears to have decreased (Figure 2B). Possible explanations for this phenomenon include increased hygiene measures at schools, more restrictive consultations and avoidance of the emergency room for fear of infection with SARS-CoV-2. The fact that the number of children admitted to the ICU did not change significantly indicates that children received adequate medical care at the time of our study. However, we did not take into account the potential long term psychiatric and social negative side-effects of the pandemic on children. 

The median hospital stay duration did not change during the time period observed in our study. At no point in our study was the ICU capacity at its limit due to an increase of patients in contrast to ICUs specialized in adult medical care. The most common clinical symptoms of our SARS-CoV-2 positive patients were fever, cough, dyspnea or tachypnea and rhinitis. In our study hospitalized SARS-CoV-2 patients did not need any form of therapy escalation. The most common drug agent administered was bronchodilator medication. Severe clinical courses were not present in our study cohort.

The relative lymphocytopenia and only moderately elevated CRP observed in infected children in our cohort are comparable to the findings of other studies [12]. 

During the pandemic the German government tried to reduce COVID-19 morbidity and mortality and maintain adequate healthcare capacities for the treatment of severely affected patients. It is therefore crucial to continuously develop and optimize strategies aimed at preventing nosocomial outbreaks The strategy to test only symptomatic SARS-CoV-2 patients as conducted during recruitment of cohort 2 in our study is no longer sufficient due to the fact that SARS-CoV-2 positive patients can present themselves with various clinical symptoms and at times remain asymptomatic. General screening measures for all admitted patients were implemented starting in October 2020. There were no nosocomial outbreaks of SARS-CoV-2 in our institution during the period observed in our study.

## 5. Limitations of the Study

Several limitations must be addressed. At the time of our study, genome analyses of the virus were not standard due to limited capacity and resources. Therefore, we were unable to determine whether certain virus variants were more dangerous for pediatric patients. It can be safely assumed that the virus variants in our cohort corresponded to that of the general German population at the time of the study. However, our results may not apply to future variants with other pathogenic features. The total number of SARS-CoV-2 admitted patients with clinical symptoms in our study was low and none of these patients required intensive care treatment. This is most likely due to the fact that SARS-CoV-2 infected pediatric patients frequently remain asymptomatic or only develop mild symptoms that do not require inpatient treatment. Due to this small number of symptomatic patients, the included description of clinical symptoms of Sars-CoV-2 positive patients may be difficult to generalize. However, in our view, this description is noteworthy and interesting for the scientific world as so far most other pediatric studies on SARS-CoV-2 have limited numbers of symptomatic patients for the same reason. 

According to the test strategy in our clinical center, most symptomatic patients presented to the hospital or were incidentally found on screening. Thus, the numbers of SARS-CoV-2 positive patients in our study cohort may underrepresent the true incidence in the general population. However, we could show that positive SARS-CoV-2 PCR results in our study population clustered with the increased overall incidence but were not limited to that (Figure 2A). Our study is a cross-sectional study and therefore cause and effect analyses cannot be performed.

## 6. Conclusions

Our study yields similar results to previous reports that SARS-CoV-2 infected children frequently remain asymptomatic or develop only mild clinical symptoms with no need for invasive therapy. Regardless, the coronavirus pandemic is an exceptional challenge for children’s health. In addition to the risk of SARS-CoV-2 infection, secondary burdens such as social isolation, conflicts within the family, decreased physical activity, screen exposure (including social-media use) or poorly balanced diets high in ultra-processed food may have multiple short- and long-term effects. Further studies are required to analyze the long-term side effects the pandemic and lockdown measures have had on the psychiatric, social and physical well-being of children. 

It is noteworthy that the number of hospital admissions apparently went down during the pandemic. This parallels observations in adult care, where utilization of the healthcare system even went down for life-threatening conditions such as myocardial infarctions or oncological diseases. One may worry that children requiring treatment did not receive adequate care during the pandemic. In this context, the more liberal handling of the mandatory examinations for early diagnosis in childhood “U1-9” in Germany may be critically discussed. 

The health and economic long-term effects have to be taken into account and practical consequences of the low frequency of severe COVID-19 in children but at the same time potential negative secondary effects on children’s health to be discussed within health care society and politics.

## Figures and Tables

**Figure 1 healthcare-09-01190-f001:**
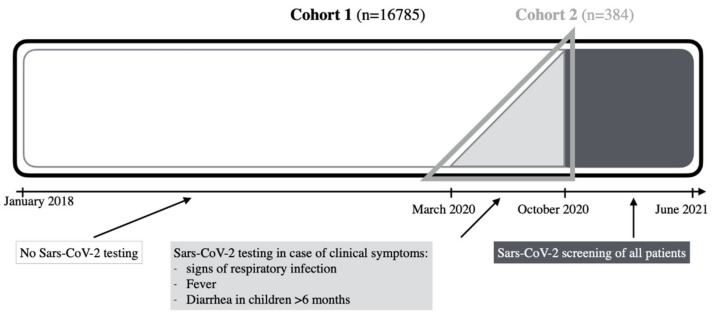
Overview of 2 distinct cohorts of pediatric patients admitted to our clinical center in our study. The retrospective analyses involved 16,785 pediatric patients hospitalized in our center between January 2018 and June 2021 (cohort 1, *n* = 16785). SARS-CoV-2 testing was implemented after onset of the pandemic in January 2020, the first test in our clinic was carried out in March 2020. Until October 2020, SARS-CoV-2 tests were performed upon clinical suspicion only (light grey area, *n* = 384), patients tested during these months were specified as cohort 2. From October 2020, all hospitalized patients were screened for SARS-CoV-2 (dark grey area, *n* = 2130).

**Figure 2 healthcare-09-01190-f002:**
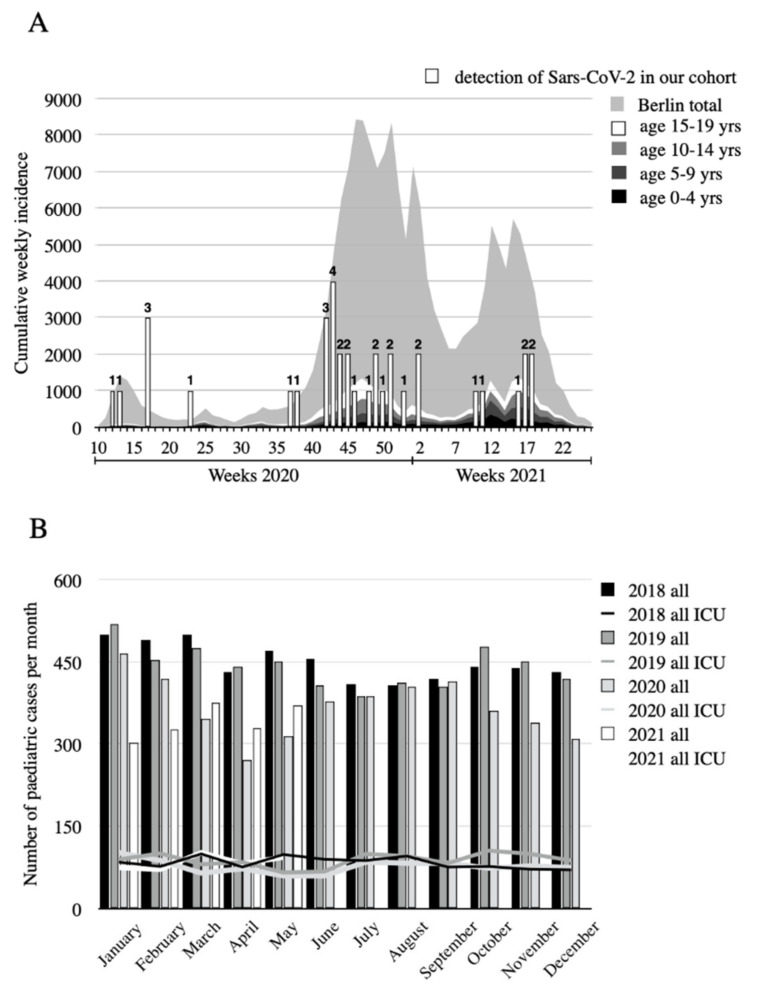
Assortment of positive SARS-CoV-2 tests in our cohort with the overall dynamic of the pandemic (**A**) and compilation of pediatric case numbers and intensive care unit (ICU) cases 2018–2021 (**B**). Local SARS-CoV-2 incidence per calendar week at our study site and area served in Berlin, Germany was obtained from Robert-Koch-Institute (grey graph, age groups 0–19 years in different greyscale as indicated) and plotted together with positive SARS-CoV-2 tests in our cohort 1 (white bars, scale is 1/1000 of *x*-axis, the actual number of positive tests is given above) (**A**). Total monthly number of pediatric hospital admissions is shown in the bar graph, the number of children admitted to ICU indicated with lines (black: 2018, grey: 2019; light grey: 2020; white: 2021) (**B**).

**Figure 3 healthcare-09-01190-f003:**
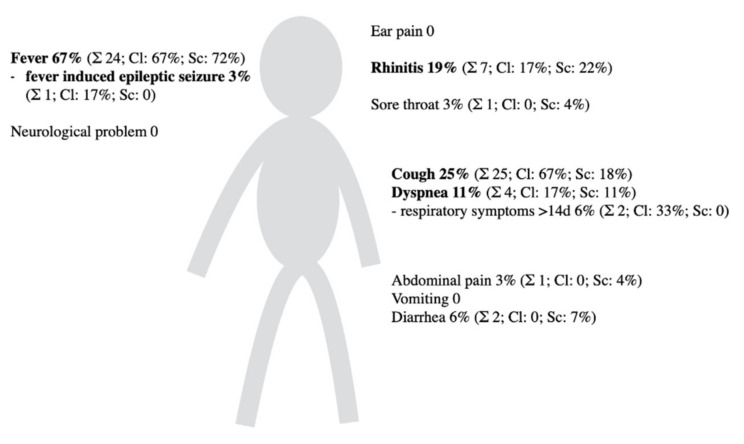
Clinical signs of SARS-CoV-2 infection in children in our cohorts. Numbers denote percentage of children from all age groups with the symptom among all infected children from cohort 1. In brackets, the total number of children with the clinical sign is given together with the percentage among positively tested during symptom-based screening of Cohort 2 March until September 2020 (total six) and unbiased screening since October 2020 (total 28).

**Table 1 healthcare-09-01190-t001:** Pediatric hospital admissions, SARS-CoV-2 test numbers and results and clinical symptoms compiled together with the approximate test positive rate in Germany (all age groups and test sites, source: Robert-Koch-Institute).

		Pediatric Hospital Admissions	SARS-CoV-2 Tests (Total)	SARS-CoV-2 Positive (Total; %)		Among These: Typical Clinical Signs Present as Previously Defined (Total; %)		Approximate Test Positive Rate in Germany (all Age Groups)
January	2020	466						
February	2020	418	0					
March	2020	345	2	2	100.0%	2	100.0%	7.7%
April	2020	271	4	3	75.0%	1	33.3%	6.6%
May	2020	314	25	0	0.0%			2.1%
June	2020	377	100	1	1.0%	1	100.0%	0.9%
July	2020	388	118	0	0.0%			0.7%
August	2020	404	87	0	0.0%			0.9%
September	2020	414	112	2	1.8%	2	100.0%	1.2%
October	2020	361	190	9	4.7%	5	55.6%	4.4%
November	2020	339	263	5	1.9%	3	60.0%	8.4%
December	2020	310	246	5	2.0%	4	80.0%	11.8%
January	2021	303	242	1	0.4%	1	100.0%	9.9%
February	2021	326	270	1	0.4%	0	0.0%	6.3%
March	2021	374	295	2	0.7%	2	100,0%	9.3%
April	2021	328	259	3	1.2%	2	66.7%	11.7%
May	2021	370	299	2	0.7%	2	100.0%	6.6%

**Table 2 healthcare-09-01190-t002:** Basic characteristics of patients admitted to the pediatric department with clinical suspicion for SARS-CoV-2 infection defined by fever and/or upper respiratory tract infection and/or diarrhea in children ≤6 months.

	SARS-CoV-2 Negative (*n* = 378)	SARS-CoV-2 Positive (*n* = 6)	Σ	SARS-CoV-2 Positive
male	219	4	223	1.8%
female	159	2	161	1.2%
age 0–6 months	96	2	98	2.0%
age 7–24 months	127	2	129	1.6%
age 2–12 years	119	1	120	0.8%
age >12 years	36	1	37	2.7%
comorbidities present	121	1	122	0.8%
neurological	15		15	
psychiatric (e.g., ADHS, depression)	1		1	
pulmonary (e.g., Asthma bronchiale, BPD, previous obstructive bronchitis or pneumonia)	53		53	
congenital heart defect	7		7	
urogenital	7		7	
syndromal (e.g., trisomy 21)	9		9	
preterm birth				
- <30 weeks of gestation	7		7	
- >30 weeks of gestation	15	1	16	6.3%
weight percentile				
<3	10	1	11	9.1%
3–9	25		25	
10–24	56		56	
25–74	184	2	186	1.1%
75–89	59	1	60	1.7%
90–96	20		20	
>97	20	1	21	4.8%
no information	4	1	5	20.0%
height percentile				
<3	15	1	16	6.3%
3–9	23		23	
10–24	44		44	
25–74	169	2	171	1.2%
75–89	54	1	55	1.8%
90–96	38	1	39	2.6%
>97	20		20	
no information	15	1	16	6.3%

**Table 3 healthcare-09-01190-t003:** Initial clinical presentation of children tested for SARS-CoV-2 upon clinical suspicion in different age groups (age in months [mo] or years [yrs] (Cohort 2).

	SARS-CoV-2 Negative (*n* = 378)				SARS-CoV-2 Positive (*n* = 6)					
	0–6 Months	7–24 Months	2–12 Years	>12 Years	0–6 Months	7–24 Months	2–12 Years	>12 Years	Σ	SARS-CoV-2 Positive
fever	57	101	67	16	2	1		1	245	1.63%
cough	30	39	70	5	1	1	1	1	148	2.7%
dyspnea or tachypnea	26	46	61	11		1			145	0.7%
- respiratory symptoms >2 weeks	2		1	1		1	1		6	33.3%
rhinitis	29	46	43	2		1			121	0.8%
sore throat	1	7	17	4					29	
headache			5	10					15	
diarrhea	20	20	12	5					57	
vomiting	6	14	26	5					51	
epileptic seizure	4	20	9	2		1			36	2.8%
abdominal pain	2	4	15	8					29	
ear pain	2	2	1	1					6	
neurological problems	2	1	4	1					8	

**Table 4 healthcare-09-01190-t004:** Final diagnosis of patients admitted to the pediatric department with clinical suspicion for SARS-CoV-2 infection but negative SARS-CoV-2 PCR from oropharyngeal swab. The percentage refers to total number of children with the respective diagnosis in all age groups. Some patients carried more than one infectious diagnosis while some other patients were categorized to non-infectious diagnoses only. Age is represented in months (mo) or years (yrs).

	0–6 Months		7–24 Months		2–12 Years		>12 Years	
upper airway infection	28	32.6%	30	21.4%	8	7.2%	5	31.3%
obstructive bronchitis	10	11.6%	33	23.6%	50	45.0%	1	6.3%
pseudocroup	2	2.3%	4	2.9%	2	1.8%		
pneumonia	2	2.3%	13	9.3%	18	16.2%	3	18.8%
- radiological diagnosis	1		4		9		1	5.3%
- clinical diagnosis	1		9		9		2	10.5%
gastroenteritis	11	12.8%	12	8.6%	15	13.5%	6	37.5%
sepsis	4	4.7%	2	1.4%				
newborn infection	10	11.6%						
febrile seizure	2	2.3%	25	17.9%	8	7.2%		
urinary tract infection	12	14.0%	10	7.1%	7	6.3%	1	6.3%
tonsillitis			7	5.0%	2	1.8%		
otitis media	3	3,5%	3	2.1%				
meningitis	2	2,3%	1	0.7%	1	0.9%		

## Data Availability

Raw data will be provided upon request by the authors.

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
