# Peer review of "SARS-CoV-2 in Pediatric Inpatient Care: Management, Clinical Presentation and Utilization of Healthcare Capacity"

_healthcare, 2021, doi:10.3390/healthcare9091190_

Round 1
Reviewer 1 Report
Thank you for the opportunity to review this paper reporting a single centre German experience of paediatric admissions during the first eighteen months, up to the end of May 2021, of the COVID-19 pandemic. The authors report on the incidence of positive COVID tests and symptomatic patients in hospitalised children, the effect of COVID-19 on the frequency and duration of inpatient treatment, and the frequency and severity of clinical symptoms among COVID-19 positive children. The paper is well written, and the research questions are answered: admission rates for general paediatrics and paediatric surgery were reduced during the peak months of the pandemic, ICU admissions did not change, hospital stay remained the same, and almost a third of the children who tested positive were asymptomatic for fever, and or respiratory tract symptoms, or in those younger than six months, diarrhoea.
The authors focus on a subgroup (Cohort 2) of 383 children admitted between March 2020 and September 2020 inclusive. They underwent PCR testing based upon clinical suspicion of COVID-19 infection: fever and or respiratory tract infection, and for those under the age of six months, diarrhoea. There appear to be some discrepancies in the numbers:
- In Table 1: the 383 children underwent 448 tests. How many children underwent multiple tests, and why?
- In Table 1, eight children are reported to have tested positive from March to October, but in Table 2 and line 170, this is reduced to five.
- In figure 1A, a total of 34 positive cases are documented, while in the text, line 116, 36 patients tested positive.
The five positive cases described in Table 3 had 11 symptoms. What were the common associations of symptoms? In total, the paper reports that 36 children tested positive. The paper would be strengthened by the inclusion of their symptoms. The reasons for the admission of the 11 asymptomatic patients might also be enlightening.
The findings report a single centre and must be viewed with caution as the sample size is very small and no patient required intensive care.
Reviewer 2 Report
Overall, the authors describe the number of SARS-CoV-2 positive children across the pandemic. It is a well written manuscript however, as the incidence reported by the authors is so low (only 36 children positive out of 6000+ patients), describing clinical features can be difficult to generalize. There may be a potential bias as most symptomatic patients presented to the hospital or were incidentally found on screening and may under-represent the true incidence in general population. The authors do touch upon it briefly in the end but it is imperative for the authors to highlight that in a separate section in the manuscript with a subheading "Limitation of the Study"
Round 2
Reviewer 1 Report
Thank you for the opportunity to review the revised paper. It is disappointing to see that there are still issues with numbers reported in the paper. Repeated errors unfortunately make one begin to question the veracity of all the data.
In Figure 1, the size of Cohort 2 is given as n=383. In the figure’s legend the cohort size has been amended to 384. This is repeated in line 78 and line 169.
The additional text added, lines 179-182 also appears to contain errors. Thirty-six children tested COVID-19 positive but ages are only reported for 33 children.
Author Response
Dear Ms Luo,
Dr. Cristina Lidón-Moyano,
Dr. Adrián González-Marrón,
Dear reviewers,
On behalf of our co-authors, we hereby submit the refined version of our manuscript healthcare-1360394 “SARS-CoV-2 in paediatric inpatient care: Management, clinical presentation and utilisation of healthcare capacity”.
We scrutinized the manuscript again and corrected Figure 1 according to the very helpful comment made by one reviewer. Any revisions to the manuscript are marked up using the “TrackChanges” function. The following paragraphs show in detail how we dealt with the suggestions. For ease of communication, we have copied the reviews into this letter, and we used the following layout:
Reviewer comments are printed in Arial without indentation.
Our responses to the reviewers are also printed in Arial, but in italics.
If necessary, excerpts from the revised manuscript are printed in Times, and indented even further.
We hope that the revised version of the manuscript will be considered suitable for publication in “healthcare”.
Sincerely,
Dr. Christine Busch, Dr. Laura Katharina Sievers and PD Dr. Constanze Pfitzer
Reviewer 1
Comments and Suggestions for Authors
Thank you for the opportunity to review the revised paper. It is disappointing to see that there are still issues with numbers reported in the paper. Repeated errors unfortunately make one begin to question the veracity of all the data.
We thank the reviewer for the thorough review and excuse that during the major revision, two details escaped our sight! We have critically reviewed how the errors could happen and provide the original table below.
In Figure 1, the size of Cohort 2 is given as n=383. In the figure’s legend the cohort size has been amended to 384. This is repeated in line 78 and line 169.
We apologize that while correcting Figure legend and text, Figure 1 itself had not been updated in the uploaded version of the manuscript. The updated and correct Figure is incorporated to the manuscript now.
The additional text added, lines 179-182 also appears to contain errors. Thirty-six children tested COVID-19 positive but ages are only reported for 33 children.
Thank you very much for this thorough review, we appreciate your detailed assessment. Previously, the age of 3 patients had not been part because we assessed age in months and years and for these patients, the age was only given in decimals of the year. Please find the original data and the corrected text below.
Almost half of the patients were younger than 6 months (17/36), 6 patients were aged 7-24 months,
4 patients aged 2-12 years and 9 patients were older than 12 years.
